# MicroRNA750-3p Targets *Processing of Precursor 7* to Suppress Rice Black-Streaked Dwarf Virus Propagation in Vector *Laodelphax striatellus*

**DOI:** 10.3390/v16010097

**Published:** 2024-01-08

**Authors:** Haitao Wang, Yan Dong, Qiufang Xu, Man Wang, Shuo Li, Yinghua Ji

**Affiliations:** Institute of Plant Protection, Key Laboratory of Food Quality and Safety of Jiangsu Province, Jiangsu Academy of Agricultural Sciences, Nanjing 210014, China

**Keywords:** microRNA750-3p, processing of precursor 7, rice black-streaked dwarf virus, small brown planthopper, *Laodelphax striatellus*, viral propagation

## Abstract

MicroRNAs (miRNAs) are non-coding RNAs, which, as members of the RNA interference pathway, play a pivotal role in antiviral infection. Almost 80% of plant viruses are transmitted by insect vectors; however, little is known about the interaction of the miRNAs of insect vectors with plant viruses. Here, we took rice black-streaked dwarf virus (RBSDV), a devastating virus to rice production in eastern Asia, and the small brown planthopper, (SBPH, *Laodelphax striatellus*) as a model to investigate the role of microRNA750-3p (miR750-3p) in regulating viral transmission. Our results showed that Ls-miR750-3p was downregulated in RBSDV-infected SBPH and predominately expressed in the midgut of SBPH. Injection with miR750-3p agomir significantly reduced viral accumulation, and the injection with the miR750-3p inhibitor, antagomir-750-3p, dramatically promoted the viral accumulation in SBPH, as detected using RT-qPCR and Western blotting. The *processing of precursor 7* (*POP7*), a subunit of RNase P and RNase MRP, was screened, identified, and verified using a dual luciferase reporter assay as one target of miR750-3p. Knockdown of *POP7* notably increased RBSDV viral propagation in SBPH and then increased the viral transmission rate by SBPH. Taken together, our data indicate that miR750-3p targets *POP7* to suppress RBSDV infection in its insect vector. These results enriched the role of POP7 in modulating virus infection in host insects and shared new insight into the function of miRNAs in plant virus and insect vector interaction.

## 1. Introduction

Many plant viruses that cause substantial agricultural losses, such as reoviruses, tospoviruses, tenuiviruses, and rhabdoviruses, are transmitted by sap-sucking insects, including leafhoppers, planthoppers, and thrips in a persistent propagative manner [1,2,3]. Persistent propagative infection allows a virus to establish its initial infection site in the alimentary canal epithelium, then replicate, assemble, disseminate, and eventually spread into the whole body of the insect vector [3,4,5]. Virus infection in the cells of insect vectors arouses the immune system to defend against the viral infection. The best-studied insect innate immune system pathways include Toll-like receptors (Toll), immune deficiency (Imd), Janus kinase-signal transducer and activator of transcription (JAK-STAT), nuclear factor-kappa B (Nf-kB), RNA interference (RNAi) antiviral pathways, apoptosis, and autophagy [6,7]; however, how the immune system resists and modulates viral infection remains largely unknown.

RNAi pathways, including the small interfering RNA (siRNA), microRNA (miRNA), and piwi-interacting RNA (piRNA), are involved with the antiviral immune response in human and animal hosts [8]. MicroRNAs (miRNAs) are short, non-coding RNAs approximately 20–24 nucleotides long that negatively regulate gene expression through translational inhibition or cleavage of complementary mRNA [9]. The mechanisms by which miRNAs modulate virus infection could be divided into three types: (1) targeting the host genes to promote or inhibit the infection with the virus [10,11]; (2) miRNA derived from the virus targeting host genes to regulate the virus infection [12,13,14,15]; (3) host miRNAs binding to RNA virus genomes to regulate virus translation, replication, and viral pathogenesis [16,17,18]. However, the mechanism by which insect vector miRNAs modulate plant virus infection remains largely elusive.

The rice black-streaked dwarf virus (RBSDV), a double-stranded RNA reovirus in the genus *Fijivirus* of the *Reoviridae* family, is transmitted by small brown planthopper (SBPH, *Laodelphax striatellus* Fallén) in a persistent propagative manner and causes economically serious reductions in rice yields in Asia [19]. Recently, some researchers have reported the role of the miRNAs of insect vectors in plant viruses. For example, a small RNA derived from rice stripe virus (RSV) downregulates the miR263a of SBPH and then suppresses RSV infection [20]. Some researchers found that miR315-5p and miR184-3p could effectively modulate RBSDV infection in SBPH [21,22]. Li et al. analyzed the conserved and novel miRNA profiles in RBSDV-infected SBPH and found miR750-3p was downregulated in RBSDV-infected SBPH [23]. Shrimp pmo-miR750 could regulate the expression of sarcoplasmic calcium-binding protein to facilitate virus infection in *Penaeus monodon* [24]; however, the mechanism of whether and how *L. striatellus* miR750-3p (Ls-miR750-3p) modulates the infection process of RBSDV remains unclear.

Processing of precursor 7 (POP7) is one of the subunits of ribonuclease (RNase) P and RNase MRP (mitochondrial RNA processing), also known as ribonuclease P protein subunit p20 (Rpp20) in flies [25]. P3 is a protein-binding domain of RNase P and RNase MRP, interacting with POP6 and POP7 directly and specifically [26,27,28,29]. The engineered RNase P-based ribozyme variant could effectively inhibit the replication and propagation of herpes simplex virus 1 (HSV-1) and human cytomegalovirus (HCMV) [30,31]. Although there are studies on the crystal structure of POP7 interacting with the P3 domain [32], as the subunit of RNase P and RNase MRP, whether POP7 modulates plant virus infection in insect vectors remains unknown.

In this research, we investigated the role of Ls-miR750-3p in RBSDV infection progress in SBPH. We revealed that RBSDV infection decreased Ls-miR750-3p expression, and the downregulated miRNA reduced the abundance of RBSDV by targeting *LsPOP7*, the subunit of RNase P/MRP in SBPH. Knockdown of *LsPOP7* significantly reduced the accumulation of RBSDV in SBPH and thereafter suppressed the viral transmission rate by SBPH. Thus, these results provide a possible agene for virus control and shed light on the role of RNase P/MRP in plant virus infection in insect vectors.

## 2. Materials and Methods

### 2.1. Insect, Virus, and Cells

*L. striatellus*, small brown planthopper (SBPH), was originally collected from Haian in Jiangsu Province in Eastern China; the non-rice-viruses-infected rice planthopper colonies were screened and reared on rice seedlings (Wuyujing No. 3) in a controlled environment, at 26 °C with 55 ± 5% humidity and a photoperiod of (16 h light, 8 h dark. RBSDV-infected rice plants were propagated by RBSDV-infected SBPH.

*Drosophila Schneider* 2 (S2) cells (Thermo Fisher Scientific, Waltham, MA, USA) were cultured at 28 °C without CO_2_ in insect cell culture medium (HyClone, Logan, UT, USA) containing 2% fetal bovine serum (Thermo Fisher Scientific, Waltham, MA, USA) and 2% penicillin/streptomycin/amphotericin B solution (Sangon Biotech, Shanghai, China).

### 2.2. RNA Extraction and RT-qPCR Assay

The total RNAs of SBPH were extracted using RNAiso plus (Takara) according to the manufacturer’s instructions. The cDNA was reverse-transcribed from 1 μg of total RNAs extracted with PrimeScript™ RT reagent kit with gDNA Eraser (Takara, Dalian, China) following the manufacturer’s instructions. RT-qPCR was performed using IQTM 5 multicolor real-time PCR detection system (BIO-RAD, Hercules, CA, USA) with TB Green Premix Ex Taq reagent (Takara, Dalian, China). The U6 snRNA and α-tubulin gene were used as endogenous controls for miRNAs and mRNAs, respectively (primers listed in Appendix A). The results were calculated using the 2^−ΔΔCT^ (cycle threshold) method. Each experiment contained three independent biological and three technical replications.

### 2.3. Expression Profiles of Ls-miR750-3p in SBPH

The 2nd-instar SBPH nymphs were fed on RBSDV-infected rice plants for 2 days (d), then transferred to healthy rice seedlings for 6 d. The 2nd-instar SBPH nymphs reared on healthy rice seedlings were collected as controls. The expression of Ls-miR750-3p was determined using RT-qPCR assays. This experiment was carried out with 30 insects and three independent biological repetitions.

The second- to fifth-instar nymphs, and female and male adults of SBPH were collected. The whole insect, central nervous system, salivary gland, midgut, ovary, and testis of SBPH were dissected with sterile forceps under a stereomicroscope in chilled 1 × phosphate-buffered solution (PBS, pH 7.4). The relative expression of Ls-miR750-3p in different developmental stages and tissues was determined using a RT-qPCR assay. Each sample contained 30 insects, and each sample contained three independent biological repetitions.

### 2.4. Effect of Agomir-750-3p and Antagomir-750-3p on RBSDV Infection

The agomir-750-3p, antagomir-750-3p, agomir-NC, and antagomir-NC were synthesized by GenePharma (Shanghai, China). The 2nd-instar SBPH nymphs were fed on RBSDV-infected rice plants for 2 d, then the SBPH were injected with a FemtoJet microinjector (Eppendorf, Hamburg, Germany) with 100 nL of 20 μM agomir-NC, agomir-750-3p, antagomir-NC, or antagomir-750-3p, respectively. The injected nymphs were transferred to healthy rice seedlings at 4 d and 10 d after injection, and the effect of miR750-3p on RBSDV accumulation was determined using RT-qPCR and Western blotting, respectively. Each sample contained 30 planthopper nymphs, and each experiment was repeated three times.

### 2.5. SDS-PAGE and Western Blot Assay

SBPH proteins were lysed with RIPA buffer (Beyotime Biotec., Shanghai, China). All samples were suspended with 5 × loading buffer and boiled for 10 min and then detected with SDS-polyacrylamide gel electrophoresis (PAGE). The viral P10 protein in SBPH was detected with P10-specific antibodies via immunoblot assay after SDS-PAGE, followed by HRP -conjugated secondary antibody (Beyotime Biotec. Shanghai, China). The β-actin protein was detected as the loading control. Blotted membranes were visualized using a Chemiluminescence gel imaging system.

### 2.6. Dual Luciferase Reporter Assay

The miRanda (http://mirdb.org/miRDB/, accessed on 5 January 2024) TargetScan (http://www.targetscan.org/vert_61, accessed on 5 January 2024) program was used to predict the target genes of Ls-miR750-3p separately. Default parameters were used for all algorithms. The 3′-UTR interacting region of the target gene was inserted into pGL3.0-basic vector, as described previously [31]. The mutant sequence of binding sites was synthesized by Tsingke Biotech (Beijing, China) and then inserted into pGL3.0-basic vector. The S2 cells, used for luciferase assay, were transfected with target plasmids (100 ng), agomir-750-3p (120 μM), or agomir-NC (120 μM) using lipofectamine 3000 (Thermo Fisher Scientific, Waltham, MA, USA, L3000008) reagent. After 48 h, the luciferase activities were measured with a dual luciferase reporter assay system (Promega, Madison, Hong Kong, China). Each transfection was performed in triplicate.

### 2.7. RNA Interference

The *POP7* and *enhanced green fluorescent protein* (*eGFP*) gene sequences for RNAi were amplified via PCR using the primers listed in Appendix A. A T7 RioMAXTM Expression RNAi System (Promega, Madison, MA, USA) was purchased to synthesize the double-stranded RNA (dsRNA) according to the manufacturer’s instructions. After the 2nd-instar planthoppers were reared on RBSDV-infected rice plants for 2 d, the nymphs were injected with dsRNAs (100 nL, 500 ng/μL) using a FemtoJet microinjector (Eppendorf, Hamburg, Germany). The dsRNA of *eGFP* was injected as a control. The SBPH was transferred immediately to healthy rice plants after microinjection. The relative expression of the genes was quantified at 4 d after dsRNA injection. Each experiment was repeated three times.

### 2.8. RBSDV Transmission Experiment

The 2nd-instar nymphs of SBPH were fed RBSDV-infected rice plants for 2 d and then injected with ds*eGFP* or ds*POP7* (500 ng/μL). The dsRNA-injected SBPH were then transferred to healthy rice seedlings for 14 d. Then, 20 rice seedlings were prepared for the transmission rate experiment, in which each rice seedling was fed with 1 dsRNA-treated SBPH for 2 d. The rice seedlings were planted in a controlled environment for 30 d, and the RBSDV transmission rate by SBPH was calculated by detecting the number of RBSDV-positive rice plants/total number of surviving rice plants. Each experiment was repeated three times.

### 2.9. Immunofluorescence Microscopy

The midguts of SBPH were dissected at 6 d after dsRNA injection on chilled 1 × PBS buffer, fixed with 2% *v*/*v* paraformaldehyde in PBS for 2 h, and then permeabilized with 2% Triton X-100 in PBS for 30 min. The organs were then immunolabeled with RBSDV-P10-specific antibody for 1 h. All samples were analyzed with LSM 710 (ZEISS, Oberkochen, Germany) for confocal microscopy.

### 2.10. Statistical Analysis

All data in this research were analyzed using SPSS software (version 19.0). Student’s *t*-test was used to compare the differences between the treatments and the controls. Data are shown as mean ± standard error (SE) from at least three independent experiments. * *p* < 0.05, ** *p* < 0.01. One-way ANOVA followed by Tukey’s test was applied for multiple comparisons. Different letters indicate significant differences at *p* < 0.05.

## 3. Results

### 3.1. The Analysis of Ls-miR750-3p Expression Profiles in SBPH

To understand the endogenous expression of Ls-miR750-3p in SBPH, we analyzed its relative expression in RBSDV-infected (RB) and RBSDV-free (VF) SBPH, and we found that the expression of Ls-miR750-3p was dramatically reduced in the RBSDV-infected group compared with that in the RBSDV-free SBPH (Figure 1A).

To figure out the expression maps of Ls-miR750-3p in SBPH spatially and temporally, we detected the expression using a RT-qPCR assay and found that Ls-miR750-3p was expressed predominantly in the male adult SBPH compared with 2nd–5thinstar and female adult SBPH (Figure 1B). In addition, the relative Ls-miR750-3p expression in second-instar SBPH was significantly lower than that in 3rd–5th-instar and male adult SBPH, and there was no difference in the Ls-miR750-3p expression among 3rd–5th-instar SBPH (Figure 1B). What is more, the expression of Ls-miR750-3p in female adult SBPH was the lowest of all the groups (Figure 1B).

We further analyzed the Ls-miR750-3p expression in different organs, and the results showed that Ls-miR750-3p expression was higher in the intestine (midgut) than in other organs (central nervous system, salivary gland, ovary, and testis) or in the whole insect (Figure 1C). The expression of Ls-miR750-3p in the whole insect was higher than that in the salivary gland, ovary, testis, or central nervous system (CNS) (Figure 1C), and there was no difference in the Ls-miR750-3p expression among CNS, salivary gland, and ovary in SBPH (Figure 1C). Thus, these results revealed the expression patterns of Ls-miR750-3p in SBPH.

### 3.2. Ls-miR750-3p Mimics or Inhibitors Modulate RBSDV Infection in SBPH

As Ls-miR750-3p responded to RBSDV infection in SBPH, to study whether Ls-miR750-3p modulates RBSDV infection in SBPH, we injected the Ls-miR750-3p agomir (mimics) or the inhibitor miR184-3p antagomir into the planthopper, separately. We analyzed and found that Ls-miR750-3p expression significantly increased in Ls-miR750-3p-agomir-injected SBPH compared with the control treatment (Figure 2A); however, the relative abundance of RBSDV significantly reduced with Ls-miR750-3p agomir injection, as analyzed with RT-qPCR assays (Figure 2B). Additionally, the viral accumulation of RBSDV at the protein level was suppressed in Ls-miR750-3p-agomir-injected SBPH as detected at 10 d after microinjection (Figure 2C).

To investigate the role of Ls-miR750-3p in the RBSDV infection process, the inhibitor antagomir of Ls-miR750-3p was injected into SBPH after 2 d of feeding on RBSDV-infected rice plants. The results of RT-qPCR showed that antagomir injection efficiently reduced relative Ls-miR750-3p expression and significantly increased the RBSDV abundance at transcript 4 d after inhibitor injection (Figure 2D,E). Additionally, we analyzed RBSDV capsid protein P10 abundance at 10 d after antagomir injection, and the results showed that the accumulation of P10 notably increased with antagomir treatment compared with the control injection (Figure 2F). That is to say, Ls-miR750-3p mimics or inhibitors could regulate RBSDV infection in SBPH, indicating that Ls-miR750-3p restricts the accumulation of RBSDV in SBPH.

### 3.3. The Candidate Targeting Gene Screening of Ls-miR750-3p

To further investigate how Ls-miR750-3p modulates RBSDV infection in SBPH, we first predicted the candidate target genes of Ls-miR750-3p using miRanda and TargetScan programs; with four genes were selected for further analysis according to the target scan scores and free energy (DG KJ/mol), namely, *calcium/calmodulin-dependent protein kinase* (*CDPK*), *neurexin* (*Neur*), *heat shock protein 60A* (*Hsp60*), and *processing of precursor 7* (*POP7*) (Figure 3A).

To confirm whether Ls-miR750-3p could directly modulate the expression of candidate genes, the Ls-miR750-3p mimics agomir or inhibitor antagomir were injected into the third-instar planthopper nymphs. At 4 d after injection, the genes’ relative expression levels were detected and analyzed using a RT-qPCR assay. The results showed that the relative expression levels of all candidate genes significantly decreased after mimic agomir injection compared with the NC control treatment (Figure 3B). Compared with other candidate genes, only the relative expression of *POP7* significantly increased after antagomir inhibitor injection (Figure 3C). Conversely, the relative expression of *Neur* showed no significant changes after the inhibitor injection compared with the control treatments, while the expression of *CDPK* or *HsP60* was notably downregulated in antagomir-injected SBPH compared with the control (Figure 3C).

To further analyze whether the expressions of the candidate targeting genes respond to RBSDV infection, the relative expression of the genes was monitored in RBSDV-infected (RB) and RBSDV-free (VF) SBPH. The results showed that the expressions of candidate target genes *CDPK*, *Hsp60*, and *POP7* was upregulated under RBSDV infection, while the relative expression of *Neur* showed no significance in virus-infected planthoppers compared with virus-uninfected SBPH (Figure 3D). Taken together, these results suggested that *POP7* might be the target gene of Ls-miR750-3p.

### 3.4. The Interaction of Ls-miR750-3p with POP7 3′-UTR Specific Region

A fluorescent luciferase assay was carried out to verify the interaction of Ls-miR750-3p with *POP7*-3′UTR. We first predicted the interaction site of *POP7* 3′-UTR with Ls-miR750-3p; the results showed that two sites in *POP7* 3′-UTR were predicted to bind the Ls-miR750-3p, separately (Figure 4A,C). The two *POP7* 3′-UTR binding sites’ sequences and mutant sequence were cloned and inserted into the pGL3.0-basic vector, separately. The recombinant plasmids named WT1 or WT2 were co-transfected with Ls-miR750-3p mimics into S2 cells, and the mutant plasmids were transfected as controls. The luciferase activity of WT2 significantly decreased compared with that of the mutant under Ls-miR750-3p mimic treatment, while the luciferase activity of WT1 co-transfected with Ls-miR750-3p mimics showed no significant difference (Figure 4B,D). These results indicated that *POP7* is the target gene of Ls-miR750-3p.

### 3.5. Knockdown of POP7 Suppresses RBSDV Propagation in SBPH

To figure out the role of POP7 in the RBSDV infection progress, we first detected the relative expression of *POP7* in the different development stages, in female and male adult SBPH, and different organs and tissues of SBPH. We found that *POP7* expression was higher in male adult SBPH than that in 2nd–5th-instar SBPH (Figure 5A). Additionally, the results showed that the expression of *POP7* did no differ among 2nd–5th-instar and female adult SBPH (Figure 5A). The organs and tissues of SBPH were dissected, and the gene expression was monitored using RT-qPCR assays. The results showed that the relative expression of *POP7* in the ovary or testis was significantly higher than in the central nervous system, salivary gland, and midgut or the whole insect (Figure 5B), and the expression of *POP7* in the midgut was the lowest among the organs and whole insect (Figure 5B).

RBSDV infection reduced the relative mRNA expression of *POP7* in SBPH (Figure 3C). To analyze the function of POP7 in the RBSDV infection progress in SBPH, *POP7* was knocked down via RNA interference induced by ds*POP7*, and the results showed that the decrease in *POP7* expression significantly restricted the accumulation of RBSDV capsid protein (P10) at the transcript level at 4 d and protein level at 10 d after dsRNA injection compared with the control ds*eGFP* treatment (Figure 5D,E). To observe the viral infection directly, we dissected the midguts of SBPH and immunolabeled them with RBSDV-P10-specific antibody at 6 d after injection. The results exhibited that ds*POP7* treatment dramatically suppressed the infection with RBSDV in the midgut compared with the control treatment (Figure 5F). What is more, we found that the knockdown of *POP7* significantly induced viral transmission (Table 1), which indicated the pivotal role of POP7 in RBSDV infection in the vector SBPH. Thus, these results indicated that knockdown of *POP7* inhibited the infection of RBSDV, which suggests the beneficial role POP7 plays in RBSDV infection in SBPH.

## 4. Discussion

The roles of miRNAs in virus infection in host plants or animals have been studied extensively in recent years; however, the role of miRNAs in the insect vectors that modulate plant virus infection remains largely unknown. We found that the expression of Ls-miR750-3p was downregulated by RBSDV infection (Figure 1A); however, what kind of factors regulate the expression remains unclear. Some researchers found virus-derived small RNAs 3397 by targeting the promoter region of miR263a to suppress the infection of RSV and maintaining a tolerable viral titer in SBPH. In addition, a transcription factor YY1 was found to negatively regulate the transcription of miR263a [19]. Whether RBSDV-derived small RNAs modulate or some transcription factor regulates the expression of Ls-miR750-3p needs further research. miRNAs are vital to the development of hosts, and Ls-miR750-3p is expressed highly in male adult SBPH (Figure 1B), but whether Ls-miR750-3p is involved the development of male SBPH needs to be investigated. The results showed that Ls-miR750-3p is predominantly expressed in the midgut, which suggests this miRNA might be important for the absorption of rice plants’ sap and may be important to defend against the pathogens in sap.

The abundance of the main capsid protein P10 is used as an indicator of RBSDV accumulation in both insects and plants [20,33]. In this study, we analyzed Ls-miR750-3p and found that it could directly restrict RBSDV infection (Figure 2A–F). Shrimp pmo-miR750 regulates the expression of sarcoplasmic calcium-binding protein, facilitating virus infection in *P. monodon* [24], but whether sarcoplasmic calcium-binding protein in SBPH promotes RBSDV infection in SBPH needs further research. Naturally occurring miRNA-binding sites within viral genomes are generally located in the 5′- and 3′-UTR [34,35] but have recently been found in the coding regions of viral proteins [17,18]. Ls-miR750-3p interacts with the *POP7*-specific 3′-UTR sequence from 716 to 740, but does not bind another sequence from 128 to 147 to modulate RBSDV infection (Figure 4A–D). Whether the knockdown of 3′-UTR sequence from 716 to 740 promotes RBSDV infection needs further investigation.

Circular RNAs (circRNAs) are endogenous RNAs that have critical regulatory roles in numerous biological processes [36,37]. CircRNAs can function as miRNA sponges, splicing interferences, and transcription regulators [38]. Some circRNAs also play important roles in virus–host interaction; for example, SBPH circRNA2030 might interact with miR315-5p and miR184-3p to modulate RBSDV infection [39]. Recent evidence demonstrates that miR315-5p promotes RBSDV infection by downregulating a melatonin receptor in SBPH [40]. The expression of miR184-3p in SBPH was upregulated by RBSDV infection [23]. It is evident from several studies that the host miRNA profile is altered following viral infection; however, whether there exist other miRNA modulates of RBSDV infection in SBPH remains largely unknown.

POP7 has endogenous ATPase activity [41]. As ATPase is critical for the replication of various viruses, whether POP7 regulates RBSDV infection by its ATPase activity needs further investigation. In addition, some research has found that POP7 could interact with heat shock protein 27 (Hsp27) and survival motor neuron 1 (SMN1) [41,42]. As such, POP7 in SBPH might interact with such proteins to modulate infection with RBSDV. The knockdown of *POP7* notably restricted the infection with RBSDV (Figure 5D–F) in SBPH; however, whether POP7 modulates the viral infection in host insects via involvement in the RNA processing pathway and the processing of capped intron-containing pre-mRNA pathway remains unknown.

In this study, we elucidated the role of Ls-miRNA750-3p and its target gene *POP7* in RBSDV infection in SBPH. To the best of our knowledge, this is the first report that POP7 is involved in modulating virus infection in host insects. Such studies will not only advance our knowledge of the fundamental processes of Ls-miRNA750-3p involved in plant virus infection but will also help us to determine POP7 function in the progression of plant virus infection.

## Figures and Tables

**Figure 1 viruses-16-00097-f001:**
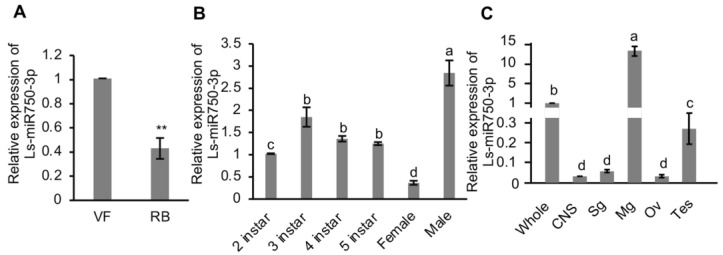
Relative expression profile of Ls-miR750-3p in SBPH. (**A**) Relative expression of Ls-miR750-3p in RBSDV-infected (RB) or RBSDV-free (VF) SBPH was examined using RT-qPCR assay. (**B**) The relative expression of Ls-miR750-3p in different development stages of female and male adults of SBPH. Each sample contained 30 SBPH. (**C**) The relative expression of Ls-miR750-3p in different organs or tissues of SBPH was detected with RT-qPCR assays. Whole, whole insect; CNS, central nervous system; Sg, salivary gland; Mg, midgut; Ov, ovary; Tes, testis. The organs or tissues of 30 SBPH were dissected for each sample. Each experiment contained three independent biological repetitions. Different letters indicate significant differences at *p* < 0.05. ** *p* < 0.01.

**Figure 2 viruses-16-00097-f002:**
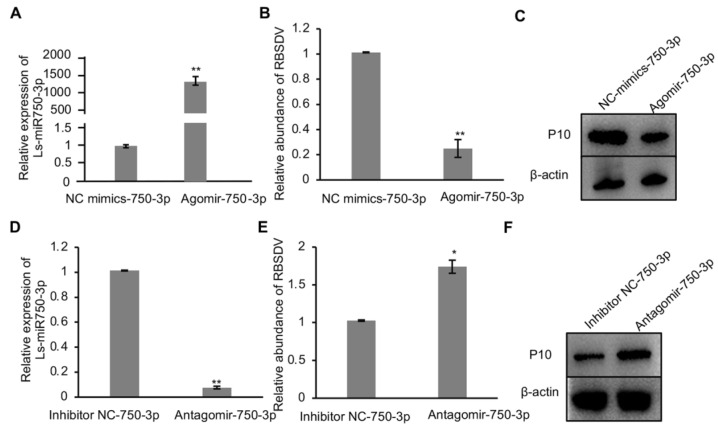
Ls-miR750-3p mimics or inhibitors regulate RBSDV infection in SBPH. (**A**) Ls-miR750-3p expression was detected via RT-qPCR 4 d after Ls-miR750-3p mimic or NC mimic injection. (**B**) The relative expression of RBSDV was detected with a RT-qPCR assay at 4 d after Ls-miR750-3p mimic or NC mimic injection. (**C**) The accumulation of RBSDV was analyzed at 10 d after mimic or NC mimic injection using Western blotting. RBSDV abundance was detected by RBSDV-P10 specific antibody. (**D**) Ls-miR750-3p expression was detected via RT-qPCR at 4 d after Ls-miR750-3p antagomir or NC inhibitor injection. (**E**) The relative expression of RBSDV-S10 was detected via RT-qPCR assay at 4 d after Ls-miR750-3p antagomir or NC inhibitor injection. (**F**) The accumulation of RBSDV-P10 was analyzed at 10 d after antagomir or NC inhibitor injection via Western blotting. RBSDV abundance was detected using RBSDV-P10-specific antibody. Each experiment contained three independent biological repetitions. * *p* < 0.05, ** *p* < 0.01.

**Figure 3 viruses-16-00097-f003:**
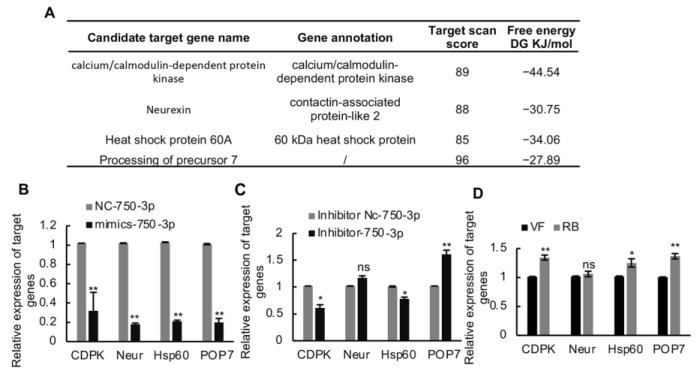
The screening of Ls-miR750-3p candidate genes. (**A**) The candidate target genes were predicted using the miRanda and TargetScan programs, and 4 target genes were selected based on the target scan score and miRanda free energy (DJ KJ/mol) for further research. (**B**) The relative expression of candidate genes was detected at 4 d after Ls-miR750-3p mimic or NC mimic injection. (**C**) The relative expression of candidate genes was detected at 4 d after Ls-miR750-3p inhibitor or NC inhibitor injection. (**D**) The relative expression of candidate genes in RBSDV-infected (8 d) and RBSDV-free SBPH (control) was detected using RT-qPCR assays: *Calcium/calmodulin-dependent protein kinase* (*CDPK*), *neurexin* (*Neur*), *heat shock protein 60A* (*Hsp60*), and *processing of precursor 7* (*POP7*). * *p* < 0.05, ** *p* < 0.01; ns, no significance.

**Figure 4 viruses-16-00097-f004:**
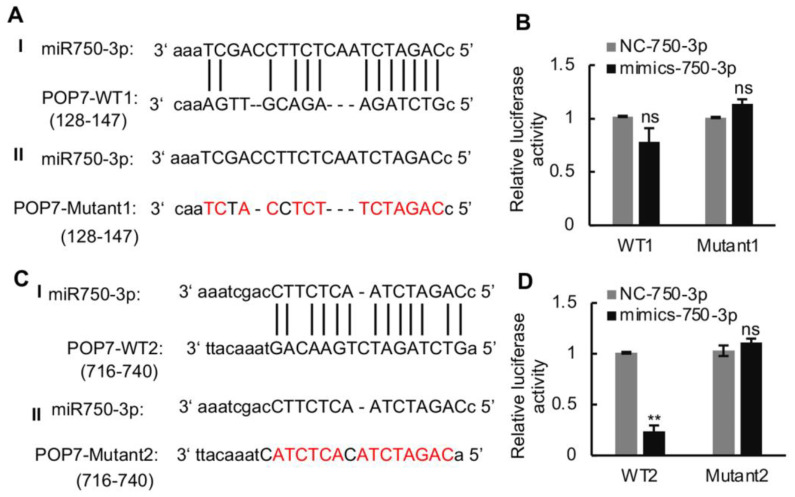
The interaction of Ls-miR750-3p with *POP7* 3′UTR was detected using a dual luciferase reporter assay. (**A**) The model of *POP7* 3′-UTR (128-147, WT1) binding site with Ls-miR750-3p. (**B**) The relative luciferase activity of WT1 with Ls-miR750-3p. (**C**) The model of *POP7* 3′-UTR (716-740, WT2) binding site with Ls-miR750-3p. (**D**) The relative luciferase activity of WT2 with Ls-miR750-3p. The plasmid *POP7*-3′UTR or *POP7*-Mut was co-transfected with mimics-miR750-3p in S2 cells, and the relative luciferase activity (pGL/pRL) was measured. NC mimics were transfected as control. The red colored letter indicates the mutant base. WT1: wild-type 1; WT2: wild-type 2. ** *p* < 0.01; ns, no significance.

**Figure 5 viruses-16-00097-f005:**
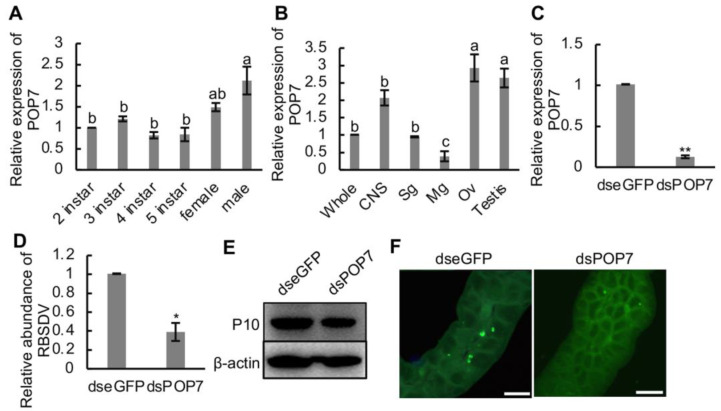
Effects of knockdown of *POP7* on RBSDV infection in SBPH. (**A**) Relative expression of *POP7* in developmental stages, and female and male adult SBPH. (**B**) Relative expression of *POP7* in different organs and tissues of SBPH. (**C**) Relative expression of *POP7* under dsRNAs treatment. (**D**) Relative expression of RBSDV abundance under dsRNAs treatment. (**E**) RBSDV accumulation in dsRNA-injected SBPH was monitored using Western blotting. RBSDV abundance was detected using RBSDV-P10-specific antibody. (**F**) Immunofluorescence microscopy showed the infection of RBSDV in dsRNA-injected SBPH. RBSDV-P10 antibody conjugated with FITC was used to detect RBSDV infection in the midgut. Different letters indicate significant differences at *p* < 0.05. * *p* < 0.05, ** *p* < 0.01. Scale bars, 20 μm.

**Table 1 viruses-16-00097-t001:** Treatment with ds*POP7* via microinjection suppressed viral transmission from insect vectors to rice plants.

Treatment ^a^	Transmission Rate (%) of RBSDV from ds*GFP*- or ds*POP7*-Injected *L. striatellus* (N = 25) to Rice Plants ^b^			*p* Value ^c^
ds*eGFP*	0.39 (7/18)	0.37 (7/19)	0.31 (5/16)	0.017
ds*POP7*	0.24 (4/17)	0.27 (4/15)	0.26 (5/19)	

^a^, Nymphs of SBPH at 2 d post-access to RBSDV-infected rice plants were injected with ds*eGFP* or ds*POP7* (500 ng/μL) and then transferred to healthy rice seedlings for 14 d. Then, 20 rice seedlings were prepared for the transmission rate experiment, where each rice seedling was fed with 1 dsRNA-treated SBPH for 2 d. ^b^, Transmission rate was calculated by detecting the number of RBSDV-positive rice plants/total number of surviving rice plants. ^c^, *p* value was estimated with Student’s *t*-test.

## Data Availability

Data are contained within the article and Appendix A.

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
