# Peer review of "MicroRNA750-3p Targets Processing of Precursor 7 to Suppress Rice Black-Streaked Dwarf Virus Propagation in Vector Laodelphax striatellus"

_viruses, 2024, doi:10.3390/v16010097_

Round 1

Reviewer 1 Report

Comments and Suggestions for Authors

In this manuscript Wang and col. report the effects that a planthopper (SBPH, small brown plant hopper, Laodelphax striatellus) microRNA (Ls-miRNA750-3p) has on rice black streaked dwarf virus (RBSDV) titers in this insect and transmission to rice plants. Specifically, the authors investigate the targeting of the Processing of Precursor 7 (POP7) by the Ls-miRNA750-3p resulting in RBSDV-transmission suppression. Previously, in another publication, it was reported that among SBPH miRNAs, the Ls-miRNA750-3p was downregulated in RBSDV-infected individuals, but the underlying mechanisms was unknown until the investigation reported in this work. To deep into the process, the authors performed a set of experiments. On one side, it was shown by RT-qPCR that Ls-miR750-3p expression was reduced in RBSDV-infected insects, confirming previous results. An agomir specific for Ls-miRNA750-3p was shown to increase its expression in injected SBPH insects, resulting in contrast, a reduction in RBSDV abundance. Conversely, the antagomir administration in the insects, reduced Ls-miRNA750-3p expression and increased RBSV abundance. Next, the authors investigated target genes for Ls-miRNA750-3p and found by computer analysis four candidates. Expression of these genes resulted affected  by the injection of the Ls-miR750-3p agomir and antagomir, showing the role of Ls-miRNA750-3p in regulating those four genes. Besides, the expression of three of the candidate genes (CDPK, Hsp60 and POP7) were reduced in RBSDV-infected insects. Next, the authors focused on investigating the interaction between Ls-miR750-3p and POP7 by experimental and computer analysis, showing the physical interaction between the microRNA and the 3’-UTR terminal end of the POP7 mRNA. In another experiment, POP7 was knockdown with a homologous dsRNA, resulting in a reduction of RBSDV accumulation in the insects and in the transmission if the virus to rice plants showing the involvement of this factor in virus accumulation and transmission. In conclusion, this manuscript provides interesting novel information on the role of insect miRNAs in virus transmission to plants.

The manuscript requires a careful edition of the English. I’ve indicated below some sentences that require modifications.

It is necessary for the authors to establish throughout the text a single nomenclature for the insect that is the subject of this work, as both the scientific name (Laodelphax striatellus and its abbreviation, L. striatellus, as well as the abbreviation of the common name, SBPH) are found indistinctly and mixed. I suggest sticking to only one, e.g. SBPH. In any case, L. striatellus should be always written in italics.

Change the sentences indicated to:

L33: Persistent-propagative viruses establish their initial …

L34: … assemble, disseminate and eventually spread …

L43:  …are involved in the antiviral….

L46: The mechanisms by which miRNAs modulate virus infection could be divided into …

L51: However, the mechanism by which insect vector miRNAs modulate plant virus infection remains largely elusive.

L61: “Penaeus monodon” in italics

L70: rewrite this sentence which is unclear.

L93: RT-qPCR was performed using the IQTM 5 multicolor real-time 93 PCR detection system…

L146: The expression of the genes was quantified at …

L194: days post microinjection

L196: To investigate the role of …

L200: Additionally, we …

L204: … , indicating that Ls-miR750-3p restricts the accumulation of RBSDV in the insect vector SBPH.

L224: the Ls-miR750-3p mimics agomir or inhibitor antagomir were injected into…

L230: Conversely, the relative expression of Neur or Hsp60 showed no significant changes after the inhibitor injection compared with the control treatments.

L234: the relative expression of the genes was monitored in RBSDV–infected (RB) …

L251: The fluorescent luciferase assay was carried out to verify …

Other comments

L129: provide references for the software used

L131: provide reference for vector pGL3.0

L181: Provide definitions of the abbreviations of the figure in the text.

Figure 3: In the Table, change Miranda energy to Free energy (DG KJ/mol). In the text for the figure it should be indicated that the DG data were obtained with the Miranda software. 

Comments on the Quality of English Language

See above comments on the English

Author Response

In this manuscript Wang and col. report the effects that a planthopper (SBPH, small brown plant hopper, Laodelphax striatellus) microRNA (Ls-miRNA750-3p) has on rice black streaked dwarf virus (RBSDV) titers in this insect and transmission to rice plants. Specifically, the authors investigate the targeting of the Processing of Precursor 7 (POP7) by the Ls-miRNA750-3p resulting in RBSDV-transmission suppression. Previously, in another publication, it was reported that among SBPH miRNAs, the Ls-miRNA750-3p was downregulated in RBSDV-infected individuals, but the underlying mechanisms was unknown until the investigation reported in this work. To deep into the process, the authors performed a set of experiments. On one side, it was shown by RT-qPCR that Ls-miR750-3p expression was reduced in RBSDV-infected insects, confirming previous results. An agomir specific for Ls-miRNA750-3p was shown to increase its expression in injected SBPH insects, resulting in contrast, a reduction in RBSDV abundance. Conversely, the antagomir administration in the insects, reduced Ls-miRNA750-3p expression and increased RBSV abundance. Next, the authors investigated target genes for Ls-miRNA750-3p and found by computer analysis four candidates. Expression of these genes resulted affected by the injection of the Ls-miR750-3p agomir and antagomir, showing the role of Ls-miRNA750-3p in regulating those four genes. Besides, the expression of three of the candidate genes (CDPK, Hsp60 and POP7) were reduced in RBSDV-infected insects. Next, the authors focused on investigating the interaction between Ls-miR750-3p and POP7 by experimental and computer analysis, showing the physical interaction between the microRNA and the 3’-UTR terminal end of the POP7 mRNA. In another experiment, POP7 was knockdown with a homologous dsRNA, resulting in a reduction of RBSDV accumulation in the insects and in the transmission if the virus to rice plants showing the involvement of this factor in virus accumulation and transmission. In conclusion, this manuscript provides interesting novel information on the role of insect miRNAs in virus transmission to plants. 

The manuscript requires a careful edition of the English. I’ve indicated below some sentences that require modifications.

It is necessary for the authors to establish throughout the text a single nomenclature for the insect that is the subject of this work, as both the scientific name (Laodelphax striatellus and its abbreviation, L. striatellus, as well as the abbreviation of the common name, SBPH) are found indistinctly and mixed. I suggest sticking to only one, e.g. SBPH. In any case, L. striatellus should be always written in italics.

Response: Thanks for your suggestion, I have changed the information in the manuscript.

Change the sentences indicated to:

L33: Persistent-propagative viruses establish their initial …

Response: Thanks, I have changed it in line 33.

L34: … assemble, disseminate and eventually spread …

Response: Thanks, I have changed it in line 34.

L43:  …are involved in the antiviral….

Response: Thanks, I have changed it in line 43.

L46: The mechanisms by which miRNAs modulate virus infection could be divided into …

Response: Thanks, I have changed it in line 47. 

L51: However, the mechanism by which insect vector miRNAs modulate plant virus infection remains largely elusive.

Response: Thanks, I have changed it in line 51.

L61: “Penaeus monodon” in italics

Response: Thanks, I have changed it in line 63-64.

L70: rewrite this sentence which is unclear.

Response: I have rewritten this sentence in line 73.

L93: RT-qPCR was performed using the IQTM 5 multicolor real-time 93 PCR detection system…

Response: I have changed the word in line 96.

L146: The expression of the genes was quantified at …

Response: I have rewritten the sentence in line 150.

L194: days post microinjection 

Response: I have rewritten the sentence in line 210.

L196: To investigate the role of …

Response: I have changed the word in line 223.

L200: Additionally, we …

Response: I have changed the word in line 227.

L204: … , indicating that Ls-miR750-3p restricts the accumulation of RBSDV in the insect vector SBPH.

Response: Thanks, I have rewritten this sentence in line 231-232.

L224: the Ls-miR750-3p mimics agomir or inhibitor antagomir were injected into… 

Response: Thanks, I have changed the word in line 241.

L230: Conversely, the relative expression of Neur or Hsp60 showed no significant changes after the inhibitor injection compared with the control treatments.

Response: Thanks, I have rewritten this sentence in line 247-250.

L234: the relative expression of the genes was monitored in RBSDV–infected (RB) …

Response: Thanks, I have changed it in line 262-263.

L251: The fluorescent luciferase assay was carried out to verify …

Response: I have changed it in line 269.

Other comments

L129: provide references for the software used

Response: The references information was provided in line 132-133.

L131: provide reference for vector pGL3.0

Response: The reference information for vector pGL3.0 were added in the manuscript in line 135.

L181: Provide definitions of the abbreviations of the figure in the text.

Response: Thanks, I have added the information of the abbreviations in the text.

Figure 3: In the Table, change Miranda energy to Free energy (DG KJ/mol). In the text for the figure it should be indicated that the DG data were obtained with the Miranda software. 

Response: Thanks, I have changed the information in the table and manuscript in line 236, 253-254.

Reviewer 2 Report

Comments and Suggestions for Authors

Haitao Wang, Yan Dong and collaborators present here a manuscript for consideration as a research article for publication in Viruses. They propose that the insect microRNA750-3p interferes with persistent-propagative infection of  RBSDV when administer exogenously and propose that POP7 RNase could be involved in the antiviral mechanism of action. The topic is interesting, and there is a lot of well-design experiments deserve some merit. However the manuscript is poorly written and lacks substantial evidence to sustain the key arguments. In my opinion, authors need to re-write the whole manuscript and improve the presentation of the data and the data analysis.

 Starting with the title:   

The down-regulated microRNA750-3p targets Processing of 2 Precursor 7 to suppress rice black streaked dwarf virus propa-3 gation in vector Laodelphax striatellus

It seems contradictory in the way is written, down-regulation of the mIR does not produces suppression of the target, at least is not what is shown in lines 160-308. Should this be change to:

>>The down-regulated microRNA750-3p targets Processing of 2 Precursor 7 to suppress rice black streaked dwarf virus propagation in the insect vector Laodelphax striatellus by down-regulating Processing of Precursor 7 POP7 subunit of RNase.

Also in the abstract in line 21. Authors sustain that “ Knockdown of LsPOP7 notably increased RBSDV viral propagation and transmission rate of SBPH” Please add the details on the transmission experiments (referred in Table 1) in the methods as a separate component.  Because is a big part of the main argument of the paper  but it is a huge jump from the prediuction of the mir target to the infected + down-regulated POP7 planthoppers, how did you call a positive plant? Is it viral infection, viral propagation or transmission rate?

Another statement that was confusing to me:

 Our results showed that Ls-15 miR750-3p was down-regulated in RBSDV infected SBPH, a highest expression level in male insect 16 and was predominately expressed in the midgut

Should this be?

>> Our results showed that Ls-15 miR750-3p was down-regulated in RBSDV infected SBPH, the highest expression level of the mir? was predominately expressed in the midgut of males insect

What the authors use to inject the mir 750-3p agomir  (the Argonatute?) or the Antagomir inhibitor, and how it is suspected to promote viral accumulation in the insect?  This needs to be expanded.

The target gene, processing of precursor 7 (LsPOP7), a sub-unit of RNase P and RNase MRP, was screened, identified and verified by the dual luciferase reporter assay.

>> was screened identified and verified for what?

Taken together, our data indicates that the miR750-3p causes LsPOP7 degradation to suppress RBSDV infection in its insect vector

Should this be?

>>Taken together, our data indicates suggests that the miR750-3p targets causes LsPOP7 degradation to and suppress RBSDV infection in its insect vector.

Line 33 : Persistent-propagative virus . This is a wrong adjective for a virus, do authors try to say  >> Persistent-propagative infection, allows viruses to establish… maybe?

In the introduction, there is a lack of information about RBSDV? Is it an RNA virus, family, replication cycle, epidemiologic consequences and most importantly how the nature of the virus impacts the molecular mechanism that implicates the mir and the RNase components.

Thus, the results will provide possible gene for virus control and shed light in the role of RNase P/MRP in the plant virus infection.

I am not sure that less virus in the insect would have an outcome in the plant infection. This needs to be tested/verified experimentally. Set up a two-arm experiment with the infected-knockout insects and infected-wt-controls, and look up for virus in the plants, how many are infected, are the infections different in terms of viral load?

Line 128- 137.  The paragraph seems to be written upside-down. Start wit the detection, goes back to adding some details about the plasmid, jumps tonto the transfection and then goes back to detection.

Additionally, Are S2 cells used for luciferase assay insect cells, please expand on the methods?

Line 162: To clarify the expression patters of the mir… >> There is nothing to be “clarified” this is your start. Should it be “To understand the endogenous expression of the miR750 in L. striatellus, we analyzed the relative expression profiles in different insect organs….”

Figure 1 needs a lot of work. All abbreviations used in the figure needs to be define in the caption. What VF and RB means? Infected and non infected? Male and female are not developmental stages as is stated in the B. portion of the caption. Define what CNS, Sg, Mg etc means.

Minor:

Line 72 – ls-miR750-3p while most of the times is refereed as Ls-miR750-3p

Line 61 – Panaeus monodon is the scientific name and needs to be italicized.

Line 86 – CO2 needs to be subscript as CO

Comments on the Quality of English Language

This manuscript needs extensive English editing before can be accepted for publication in any journal. 

Author Response

Haitao Wang, Yan Dong and collaborators present here a manuscript for consideration as a research article for publication in Viruses. They propose that the insect microRNA750-3p interferes with persistent-propagative infection of  RBSDV when administer exogenously and propose that POP7 RNase could be involved in the antiviral mechanism of action. The topic is interesting, and there is a lot of well-design experiments deserve some merit. However the manuscript is poorly written and lacks substantial evidence to sustain the key arguments. In my opinion, authors need to re-write the whole manuscript and improve the presentation of the data and the data analysis.

1.Starting with the title:   

The down-regulated microRNA750-3p targets Processing of 2 Precursor 7 to suppress rice black streaked dwarf virus propa-3 gation in vector Laodelphax striatellus

It seems contradictory in the way is written, down-regulation of the mIR does not produces suppression of the target, at least is not what is shown in lines 160-308. Should this be change to:

>>The down-regulated microRNA750-3p targets Processing of Precursor to suppress rice black streaked dwarf virus propagation in the insect vector Laodelphax striatellus by down-regulating Processing of Precursor 7 POP7 subunit of RNase.

Response: Thanks, we changed the title: The microRNA750-3p targets Processing of Precursor 7 to suppress rice black streaked dwarf virus propagation in vector Laodelphax striatellus

  1. Also in the abstract in line 21. Authors sustain that “ Knockdown of LsPOP7 notably increased RBSDV viral propagation and transmission rate of SBPH” Please add the details on the transmission experiments (referred in Table 1) in the methods as a separate component.  Because is a big part of the main argument of the paper but it is a huge jump from the prediuction of the mir target to the infected + down-regulated POP7 planthoppers, how did you call a positive plant? Is it viral infection, viral propagation or transmission rate?

 Response: Thanks, we rewritten the sentence in line 21-22. We added the details of the RBSDV transmission experiments in the materials and methods part in line 152-159.

Another statement that was confusing to me:

  1. Our results showed that Ls-15 miR750-3p was down-regulated in RBSDV infected SBPH, a highest expression level in male insect 16 and was predominately expressed in the midgut

Should this be? 

>> Our results showed that Ls-15 miR750-3p was down-regulated in RBSDV infected SBPH, thehighest expression level of the mir? was predominately expressed in the midgut of males insect

Response: Thanks, we have changed it in line 16.

  1. What the authors use to inject the mir 750-3p agomir (the Argonatute?) or the Antagomir inhibitor,

Response: We used the FemtoJet microinjector (Eppendorf, Germany) to inject the agomir-NC, agomir-750-3p, antagomir-NC, or antagomir-750-3p, respectively. We added the information in the materials and methods part in line 117. Agomir is not Argonatute, is the mimics of miRNA, and is used extensively in the miRNA research, for instance, Zhang et al., 2021; Wu et al., 2023.

  1. and how it is suspected to promote viral accumulation in the insect?  This needs to be expanded. 

Response: The RBSDV viral accumulation in SBPH was detected by RT-qPCR and western blotting, we added the information in the abstract part in line 19.

  1. The target gene, processing of precursor 7 (LsPOP7), a sub-unit of RNase P and RNase MRP, was screened, identified and verified by the dual luciferase reporter assay.

>> was screened identified and verified for what? 

Response: Thanks, we rewritten the sentence in line 19-21.

  1. Taken together, our data indicates that the miR750-3p causes LsPOP7 degradation to suppress RBSDV infection in its insect vector

Should this be? 

>>Taken together, our data indicates suggests that the miR750-3p targets causes LsPOP7 degradation to and suppress RBSDV infection in its insect vector. 

Response: We have rewritten this sentence in line 23.

Line 33 : Persistent-propagative virus . This is a wrong adjective for a virus, do authors try to say  >> 8. Persistent-propagative infection, allows viruses to establish… maybe?

Response: We have rewritten the sentence in line 33.

  1. In the introduction, there is a lack of information about RBSDV? Is it an RNA virus, family, replication cycle, epidemiologic consequences and most importantly how the nature of the virus impacts the molecular mechanism that implicates the mir and the RNase components. Thus, the results will provide possible gene for virus control and shed light in the role of RNase P/MRP in the plant virus infection.

Response: The related information was written in line 53-56, 59-60. As we know, till now, the role of RNase components in RBSDV infection remains unknown.

  1. I am not sure that less virus in the insect would have an outcome in the plant infection. This needs to be tested/verified experimentally. Set up a two-arm experiment with the infected-knockout insects and infected-wt-controls, and look up for virus in the plants, how many are infected, are the infections different in terms of viral load? 

Response: There are some researches reported that the less viral abundance caused by dsRNAs in the insect vector could significantly reduce the viral transmission rate from the insect to the plant and then result in the lower viral infection in plants. Reference: 1. Mao Q, Liao Z, Li J, Liu Y, Wu W, Chen H, Chen Q, Jia D, Wei T. Filamentous Structures Induced by a Phytoreovirus Mediate Viral Release from Salivary Glands in Its Insect Vector. J Virol. 2017, 91(12):e00265-17. doi: 10.1128/JVI.00265-17. 2. Chen Y, Chen Q, Li M, Mao Q, Chen H, Wu W, Jia D, Wei T. Autophagy pathway induced by a plant virus facilitates viral spread and transmission by its insect vector. PLoS Pathog. 2017, 13(11):e1006727. doi: 10.1371/journal.ppat.1006727. 3. Wu W, Yi G, Lv X, Mao Q, Wei T. A leafhopper saliva protein mediates horizontal transmission of viral pathogens from insect vectors into rice phloem. Commun Biol. 2022, 5(1):204. doi: 10.1038/s42003-022-03160-y.

  1. Line 128- 137.  The paragraph seems to be written upside-down. Start with the detection, goes back to adding some details about the plasmid, jumps tonto the transfection and then goes back to detection. Additionally, Are S2 cells used for luciferase assay insect cells, please expand on the methods?

Response: We first predicted the binding site of target genes with miRNA, cloned the sequence contains binding site and then inserted it into pGL3.0-basic vector for dual luciferase reporter assay. The related information for S2 cells were added in line 137.

  1. Line 162: To clarify the expression patters of the mir… >> There is nothing to be “clarified” this is your start. Should it be “To understand the endogenous expression of the miR750 in L. striatellus, we analyzed the relative expression profiles in different insect organs….”

Response: Thanks, we rewritten this sentence in line 174-175.

  1. Figure 1 needs a lot of work. All abbreviations used in the figure needs to be define in the caption. What VF and RB means? Infected and non infected? Male and female are not developmental stages as is stated in the B. portion of the caption. Define what CNS, Sg, Mgetc means.

Response: Thanks, we added the related information in the caption in line 197-200.

Minor:

  1. Line 72 – ls-miR750-3p while most of the times is refereed as Ls-miR750-3p

Response: We have changed the information in line 76.

  1. Line 61 – Panaeus monodon is the scientific name and needs to be italicized. 

Response: We have changed the information in line 63-64.

  1. Line 86 – CO2 needs to be subscript as CO2

Response: We have changed the information in line 89.

Reviewer 3 Report

Comments and Suggestions for Authors

Dear authors,

I had the pleasure to read the manuscript that starts to shed light on the role of miRNAs in insect vectors and prepare the ground to the comprehesion of the modulation of the virus infection in host insects. I found the manuscript well written and clear, step by step. Few suggestions from my side:

In the section 3.1 it is not clear how many adults were used in that experiment. When you go to the Materials and methods the authors talk about 30 insect which is not clear what they are talking about: males, females, instars etc.. That number should be also specify in all the legends the insects are, to get the meaning of the standard error. In  particular in some cases, it seems that the statistical difference is not clear (for example in Figure 5B). Please double-check all the calculation.
Please, in general, add some more information in all the legends.
100nL of dsRNA: it would be better to report the concentration
Why in Figure 3B, CDPK is ns?
The topic is fascinating and I will recommend it for publishing.
Kind regards

Author Response

Dear authors,

I had the pleasure to read the manuscript that starts to shed light on the role of miRNAs in insect vectors and prepare the ground to the comprehension of the modulation of the virus infection in host insects. I found the manuscript well written and clear, step by step. Few suggestions from my side:

In the section 3.1 it is not clear how many adults were used in that experiment. When you go to the Materials and methods the authors talk about 30 insect which is not clear what they are talking about: males, females, instars et al.

Response: Thanks, we have added the information in the materials and methods part in line 112-113.

That number should be also specified in all the legends the insects are, to get the meaning of the standard error.

Response: We have added the related information in the figure legends in line 196.

In particular in some cases, it seems that the statistical difference is not clear (for example in Figure 5B). Please double-check all the calculation. 

Response: Thanks, we recalculated the results and rewritten the results in line 295-198.
Please, in general, add some more information in all the legends.

Response: We have added the related information in the figure legends in the revised manuscript.
100nL of dsRNA: it would be better to report the concentration

Response: We have added the information (500 ng/μL) in line 148.

Why in Figure 3B, CDPK is ns?

Response: We reanalyzed the data and rewritten the results as shown in figure 3 and in text line 243-245.

Round 2

Reviewer 2 Report

Comments and Suggestions for Authors

In this 3rd revised version, the authors seem to improve the manuscript.  I will support this version to be considered for publication in Viruses. 

Comments on the Quality of English Language

Minor editing of the English language is required